# Boundary Value Problem for Weak Nonlinear Partial Differential Equations of Mixed Type with Fractional Hilfer Operator

**Tursun K. Yuldashev [1,\*] and Bakhtiyor J. Kadirkulov [2]**

[1]  Uzbek-Israel Joint Faculty of High Technology and Engineering Mathematics, National University of Uzbekistan, Tashkent 100174, Uzbekistan

[2]  Tashkent State Institute of Oriental Studies, Tashkent 100060, Uzbekistan; kadirkulovbj@gmail.com

\*  Correspondence: t.yuldashev@nuu.uz or tursun.k.yuldashev@gmail.com; Tel.: +998-99-519-59-31

**Abstract:** In this paper, we consider a boundary value problem for a nonlinear partial differential equation of mixed type with Hilfer operator of fractional integro-differentiation in a positive rectangular domain and with spectral parameter in a negative rectangular domain. With respect to the first variable, this equation is a nonlinear fractional differential equation in the positive part of the considering segment and is a second-order nonlinear differential equation with spectral parameter in the negative part of this segment. Using the Fourier series method, the solutions of nonlinear boundary value problems are constructed in the form of a Fourier series. Theorems on the existence and uniqueness of the classical solution of the problem are proved for regular values of the spectral parameter. For irregular values of the spectral parameter, an infinite number of solutions of the mixed equation in the form of a Fourier series are constructed.

**Keywords:** mixed type nonlinear equation; boundary value problem; hilfer operator; mittag–leffler function; spectral parameter; solvability

## 1. Introduction

One of the most striking areas of mathematical analysis is the invention of fractional-order integro-differential operators. Today, the theory and application of operators of fractional differentiation and integration have become a powerful industry of theoretical and applied research at the highest levels of different science and technology. In particular, a concrete physical and engineering interpretation of the generalized fractional operator is given in [1] (Volume 4–8), [2–6]. At present, the operators of fractional differentiation and integration are also widely used in the study of problems associated with the study of the coronavirus COVID-19 (see, for example [1,7]).

In this paper we use Hilfer operator:

$$D^{\alpha,\gamma} = J_{0+}^{\gamma-\alpha} \frac{d}{dt} J_{0+}^{1-\gamma}, \, 0 < \alpha \le \gamma \le 1,$$

where

$$J_{0+}^{\alpha} \varphi(t) = \frac{1}{\Gamma(\alpha)} \int_0^t \frac{\varphi(\tau) \, d\tau}{(t-\tau)^{1-\alpha}}, \; \alpha > 0$$

is a Riemann–Liouville integral operator.

For $\gamma = \alpha$ and $\gamma = 1$ we have $D^{\alpha,0} = {}_{RL}D_{0+}^{\alpha}$ and $D^{\alpha,1} = {}_{C}D_{0+}^{\alpha}$. Therefore, the generalized integro-differentiation operator $D^{\alpha,\gamma}$ is a continuous interpolation of the well-known

fractional order differentiation operators of Riemann–Liouville and Caputo, which describe diffusion processes [1] (Volume 1, pp. 47–85).

Now we consider in detail a review of some works. For the first time, the generalized Riemann–Liouville operator (named as the Hilfer fractional derivative) was introduced by R. Hilfer on the basis of fractional time evolutions that arise during the transition from the microscopic scale to the macroscopic time scale [8]. Furthermore, R. Hilfer solved a Cauchy type problem for a fractional order equation with the same operator, applying in this case the Laplace transforms. In addition, using the integral Fourier, Laplace, and Mellin transforms, he investigated the Cauchy problem for the generalized diffusion equation, the solution of which is presented in the form of the Fox H-function.

It is applied in [9,10], the generalized fractional integro-differentiation operator in studying the dielectric relaxation in glass-forming liquids with different chemical compositions. For this, as usual, a classical Debye-type model was used, which describes exponential relaxation. The Debye-type model is determined by a first-order differential equation (see Equation (19) in [9]). But, as follows from the experiments, the ubiquitous feature of the dynamics of supercooled liquids and amorphous polymers is just non-exponential relaxation, which is the result of slow relaxation. To successfully describe the relaxation dynamics of glassy materials, the author of this article proposed a new model of dielectric relaxation containing derivatives and integrals of the non-integer order, which are a natural generalization of the Debye equation.

In [11] boundary value problems for the fractional diffusion equation with the time-generalized Riemann–Liouville fractional derivative (named as the Hilfer fractional derivative) in finite and infinite domains are studied. In the finite domain, the method of separation of variables and the Laplace transform method for solving the problem were used. In addition, the solution of the considered problem was obtained in the form of an infinite series containing the Mittag–Leffler function, and the asymptotic behavior of this solution at infinity was also found. In the infinite domain with respect to the spatial variable by the Fourier-Laplace transform method, the Cauchy problem is solved. In particular, a fundamental solution of the Cauchy problem is found and the fractional moments of the fundamental solution of the fractional diffusion equation are calculated. It is also shown in [11] that the corresponding solutions of the diffusion equations with fractional derivatives in the sense of Caputo or Riemann–Liouville are particular cases of diffusion equations with a fractional derivative according to Hilfer. The results obtained in this work are relevant in the study of the dielectric relaxation of glass and problems of the aquifer.

In [12] the analytical and numerical solution of boundary value problems for the fractional diffusion equation with the Hilfer fractional derivative was studied with respect to time and with respect to the Riesz–Feller spatial fractional derivative. To solve the problem, the Laplace and Fourier transform methods were used, and the solutions are presented by the Mittag–Leffler functions and the Fox H-function. A numerical solution of the problem is also considered by aid of approximating fractional derivatives with fractional derivatives of the Grunwald–Letnikov.

In [13], a new definition of the fractional derivative is introduced: The Hilfer–Prabhakar fractional derivative, which generalizes the fractional derivatives of Riemann–Liouville and Caputo. The new operator is constructed by replacing the Riemann–Liouville integrals of fractional order with more general Prabhakar integrals of fractional order. In addition, some applications of these generalized fractional derivatives in solving classical equations of mathematical physics are shown. Here we can note the heat equations and differential-difference equations that determine the dynamics of generalized random recovery processes, etc.

In [14] the properties of the Hilfer operator were investigated in a special functional space, and an operational method was developed for solving fractional differential equations with this operator. Developing the results of [14], the authors of [15] developed an operational method for solving fractional differential equations containing a finite linear combination of Hilfer operators with various parameters.

More detailed information as well as a bibliography related to the Hilfer fractional derivative can be found in the recently published monograph [16], where the theory of fractional integro-differentiation, including the Hilfer fractional derivative, is systematically presented. Section 2 of this paper gives the basic properties of the Hilfer operators, and its generalization is the Hilfer–Prabhakar fractional derivative, and Section 4 shows the applications of these fractional derivatives in solving various applied problems of mathematical physics.

So, a large number of scientific papers have been devoted to the investigation of initial, boundary, and inverse value problems for linear and nonlinear ordinary and partial differential equations (see also [17–26]).

We note that in [27] the problem of source identification was studied for the generalized diffusion equation with operator $D^{\alpha,\gamma}$. In the work [28] the inverse problems are investigated for a generalized fourth-order parabolic equation with the operator $D^{\alpha,\gamma}$.

In nature and in physics, processes that occur over time are usually nonlinear. Therefore, the study of nonlinear differential and functional-differential equations of fractional order is relevant.

## 2. Problem Statement

In a domain $\Omega = \{-a < t < b, \ 0 < x, y < l\}$ we consider a nonlinear partial fractional differential equation of mixed type:

$$0 = \begin{cases} \left( D^{\alpha,\gamma} - D^{\alpha,\gamma}\left(\frac{\partial^2}{\partial x^2} + \frac{\partial^2}{\partial y^2}\right) - \left(\frac{\partial^2}{\partial x^2} + \frac{\partial^2}{\partial y^2}\right)\right) U\left(t, x, y\right) \\ -g_1\left(t\right) f_1\left(x, y, \int\limits_0^b \int\limits_0^l \int\limits_0^l \Theta_1\left(\theta, \zeta, \varsigma, \theta^{1-\gamma} U\left(\theta, \zeta, \varsigma\right)\right) d\theta \, d\zeta \, d\varsigma\right), \quad (t, x, y) \in \Omega_1, \\ \left( \frac{\partial^2}{\partial t^2} - \frac{\partial^2}{\partial t^2}\left(\frac{\partial^2}{\partial x^2} + \frac{\partial^2}{\partial y^2}\right) - \omega^2\left(\frac{\partial^2}{\partial x^2} + \frac{\partial^2}{\partial y^2}\right)\right) U\left(t, x, y\right) \\ -g_2\left(t\right) f_2\left(x, y, \int\limits_{-a}^0 \int\limits_0^l \int\limits_0^l \Theta_2\left(\theta, \zeta, \varsigma, U\left(\theta, \zeta, \varsigma\right)\right) d\theta \, d\zeta \, d\varsigma\right), \quad (t, x, y) \in \Omega_2, \end{cases} \tag{1}$$

where $\Omega_1 = \{0 < t < b, \ 0 < x, y < l\}$, $\Omega_2 = \{-a < t < 0, \ 0 < x, y < l\}$, $\omega$ is positive spectral parameter, and $a$, $b$ are positive real numbers,

$$D^{\alpha,\gamma} = J_{0+}^{\gamma-\alpha}\frac{d}{dt}J_{0+}^{1-\gamma}, \ 0 < \alpha \le \gamma \le 1$$

is Hilfer operator, $g_1\left(t\right) \in C\left[0\,;b\right]$, $g_2\left(t\right) \in C\left[-a\,;0\right]$,

$$f_i\left(x, y, u\right) \in C\left(\left[0;l\right]^2 \times \mathbb{R}\right), i = 1, 2, \ \Theta_1(t, x, y, U) \in C\left(\left[0;b\right] \times \left[0;l\right]^2 \times \mathbb{R}\right),$$

$$\Theta_2(t, x, y, U) \in C\left(\left[-a;0\right] \times \left[0;l\right]^2 \times \mathbb{R}\right).$$

**Problem 1** ($T_\omega$). *It is required to find a function* $U\left(t, x, y\right)$, *which belongs to the class:*

$$\begin{bmatrix} t^{1-\gamma}\frac{\partial^k U}{\partial x^k} \in C\left(\overline{\Omega}_1\right), \ t^{1-\gamma}\frac{\partial^k U}{\partial y^k} \in C\left(\overline{\Omega}_1\right), \ \frac{\partial^k U}{\partial x^k} \in C\left(\overline{\Omega}_2\right), \ \frac{\partial^k U}{\partial y^k} \in C\left(\overline{\Omega}_2\right), \\ D^{\alpha,\gamma}U \in C\left(\Omega_1\right), \ U_{tt}, \ U_{xx}, \ U_{yy} \in C\left(\Omega_1 \cup \Omega_2\right), \ k = 0, 1, 2, \end{bmatrix} \tag{2}$$

*satisfies mixed differential Equation* (1) *in the domain* $\Omega_1 \cup \Omega_2$, *boundary value conditions:*

$$U\left(t, 0, y\right) = U\left(t, l, y\right) = U\left(t, x, 0\right) = U\left(t, x, l\right) = 0, \quad t \neq 0, \tag{3}$$

$$U\left(-a, x, y\right) = U\left(b, x, y\right) + \varphi\left(x, y\right), \quad 0 \le x, y \le l, \tag{4}$$

*gluing conditions:*

$$\lim_{t\to+0} J_{0+}^{1-\gamma} U(t, x, y) = \lim_{t\to-0} U(t, x, y), \quad \lim_{t\to+0} J_{0+}^{1-\alpha} \frac{d}{dt} J_{0+}^{1-\gamma} U(t, x, y) = \lim_{t\to-0} \frac{d}{dt} U(t, x, y), \quad (5)$$

*where $\varphi(x, y)$ is given a sufficiently smooth function.*

Note that boundary value conditions of type (3) take place in modeling problems of the flow around a profile by a subsonic velocity stream with a supersonic zone. Nonlocal boundary value problems for different type of equations were studied in the works of many authors, in particular, in [29–36]. Nonlinear differential and integro-differential equations without mixing of the type of equations were studied in [37–42] by the Fourier series method.

In our work, unlike mixed parabolic-hyperbolic equations, the problem of small denominators do not arise. In this paper, we consider a boundary value problem for a mixed type nonlinear differential equation with Hilfer operator of fractional integro-differentiation. The Fourier method of separation of variables is used taking into account the features of the fractional integro-differentiation operator and nonlinearity. We study the solvability of problem (1)–(5) for various values of the spectral parameter. This work is a further development of the results of [35,38–40,42–45].

## 3. Nonhomogeneous Ordinary Differential Equation With Hilfer Operator

We consider the Cauchy problem for a nonhomogeneous differential equation of fractional order:

$$\begin{cases} D^{\alpha,\gamma} u(t) = k u(t) + f(t), & t \in (0, t_1), \\ \lim_{t\to+0} J_{0+}^{1-\gamma} u(t) = u_0, \end{cases} \quad (6)$$

where $f(t)$ is given continuous function, $u_0 = $ const.

Note that in [28], the Laplace method was applied to solve this problem. In [15], a solution was found using operational calculus for a more general problem than (6) in a specially constructed functional space. In our work, we use a more rational way to solve problem (6), which allows us to obtain an explicit solution.

We prove that there holds the following lemma.

**Lemma 1.** *Let be $f(t) \in C(0; t_1] \cap L_1(0; t_1)$. Then the solution of the problem (6) $u(t) \in C(0; t_1] \cap L_1(0; t_1)$ is represented as follows:*

$$u(t) = u_0 t^{\gamma-1} E_{\alpha,\gamma}(k t^\alpha) + \int_0^t (t-\tau)^{\alpha-1} E_{\alpha,\alpha}(k(t-\tau)^\alpha) f(\tau) d\tau, \quad (7)$$

*where*

$$E_{\alpha,\gamma}(z) = \sum_{m=0}^{\infty} \frac{z^m}{\Gamma(\alpha m + \gamma)}, \quad z, \alpha, \gamma \in \mathbb{C}, \ \ Re(\alpha) > 0$$

*is a Mittag–Leffler function (Volume 1, pp. 269–295) in [1].*

**Proof.** We rewrite the differential equation of problem (6) in the form:

$$J_{0+}^{\gamma-\alpha} D_{0+}^{\gamma} u(t) = k u(t) + f(t).$$

Applying the operator $J_{0+}^{\alpha}$ to both sides of this equation, taking into account the linearity of this operator and the following formula [15]:

$$J_{0+}^{\gamma} D_{0+}^{\gamma} u(t) = u(t) - \frac{1}{\Gamma(\gamma)} J_{0+}^{1-\gamma} u(t)|_{t=0} t^{\gamma-1},$$

we obtain:

$$u(t) = \frac{u_0}{\Gamma(\gamma)} t^{\gamma-1} + J_{0+}^{\alpha} f(t) + k J_{0+}^{\alpha} u(t). \tag{8}$$

Using the lemma from [44], we represent the solution of Equation (8) as follows:

$$u(t) = \frac{u_0}{\Gamma(\gamma)} t^{\gamma-1} + J_{0+}^{\alpha} f(t) +$$

$$+ k \int_0^t (t-\tau)^{\alpha-1} E_{\alpha,\alpha} \left(k(t-\tau)^{\alpha}\right) \left[\frac{u_0}{\Gamma(\gamma)} \tau^{\gamma-1} + J_{0+}^{\alpha} f(\tau)\right] d\tau. \tag{9}$$

We rewrite the representation (9) as the sum of two expressions:

$$I_1(t) = u_0 \left[\frac{t^{\gamma-1}}{\Gamma(\gamma)} + \frac{k}{\Gamma(\gamma)} \int_0^t (t-\tau)^{\alpha-1} E_{\alpha,\alpha} \left(k(t-\tau)^{\alpha}\right) \tau^{\gamma-1} d\tau\right], \tag{10}$$

$$I_2(t) = J_{0+}^{\alpha} f(t) + k \int_0^t (t-\tau)^{\alpha-1} E_{\alpha,\alpha} \left(k(t-\tau)^{\alpha}\right) J_{0+}^{\alpha} f(\tau) d\tau. \tag{11}$$

We apply the following representations (Volume 1, pp. 269–295) in [1]:

$$E_{\alpha,\gamma}(z) = \frac{1}{\Gamma(\gamma)} + z E_{\alpha,\gamma+\alpha}(z), \quad \alpha > 0, \quad \gamma > 0, \tag{12}$$

$$\frac{1}{\Gamma(\tau)} \int_0^z (z-t)^{\tau-1} E_{\alpha,\gamma}(kt^{\alpha}) t^{\gamma-1} dt = z^{\gamma+\tau-1} E_{\alpha,\gamma+\tau}(kz^{\alpha}), \quad \tau > 0, \quad \gamma > 0. \tag{13}$$

Then for the integral (10) we obtain:

$$I_1(t) = u_0 t^{\gamma-1} E_{\alpha,\gamma}(kt^{\alpha}). \tag{14}$$

The integral in (11) we can transform as follows:

$$\int_0^t (t-\xi)^{\alpha-1} E_{\alpha,\alpha} \left(k(t-\xi)^{\alpha}\right) J_{0+}^{\alpha} f(\xi) d\xi$$

$$= \frac{1}{\Gamma(\alpha)} \int_0^t (t-\xi)^{\alpha-1} E_{\alpha,\alpha} \left(k(t-\xi)^{\alpha}\right) d\xi \int_0^{\tau} (\xi-s)^{\alpha-1} f(s) ds$$

$$= \frac{1}{\Gamma(\alpha)} \int_0^t f(s) ds \int_s^t (t-\xi)^{\alpha-1} (\xi-s)^{\alpha-1} E_{\alpha,\alpha} \left(k(t-\xi)^{\alpha}\right) d\xi. \tag{15}$$

Taking (13) into account the second integral in the last equality of (15) can be written as:

$$\int_s^t (t-\xi)^{\alpha-1} (\xi-s)^{\alpha-1} E_{\alpha,\alpha} \left(k(t-\xi)^{\alpha}\right) d\xi = \Gamma(\alpha) (t-\xi)^{2\alpha-1} E_{\alpha,2\alpha} \left(k(t-\xi)^{\alpha}\right).$$

Then, taking into account (12), we represent (11) in the following form:

$$I_2(t) = \int_0^t (t - \xi)^{\alpha - 1} E_{\alpha, \alpha} \left( k (t - \xi)^\alpha \right) f(\xi) \, d\xi. \tag{16}$$

Substituting (14) and (16) into the sum $u(t) = I_1(t) + I_2(t)$, we obtain (7). The Lemma 1 is proved. $\square$

## 4. Formal Expansion of the Solution of the Problem (1)–(5) into Fourier Series

The solution of the mixed differential Equation (1) in the domain $\Omega$ is sought in the form of a Fourier series:

$$U(t, x, y) = \sum_{n, m=1}^{\infty} u_{n, m}(t) \, \vartheta_{n, m}(x, y), \tag{17}$$

where

$$u_{n, m}(t) = \int_0^l \int_0^l U(t, x, y) \, \vartheta_{n, m}(x, y) \, dx \, dy, \tag{18}$$

$$\vartheta_{n, m}(x, y) = \frac{2}{l} \sin(\mu_n x) \sin(\mu_m x), \quad \mu_n = \frac{n\pi}{l}, \quad \mu_m = \frac{m\pi}{l}, \quad n, m \in \mathbb{N}.$$

We suppose also that:

$$f_i(x, y, \cdot) = \sum_{n, m=1}^{\infty} f_{i\,n, m}(\cdot) \, \vartheta_{n, m}(x, y), \quad i = 1, 2, \tag{19}$$

where

$$f_{i\,n, m}(\cdot) = \int_0^l \int_0^l f_i(x, y, \cdot) \, \vartheta_{n, m}(x, y) \, dx \, dy, \quad i = 1, 2.$$

Substituting series (17) and (19) into mixed Equation (1), we obtain a countable system of differential equations:

$$D^{\alpha, \gamma} u_{n, m}(t) + \lambda_{n, m}^2 u_{n, m}(t) = g_1(t) f_{1\,n, m}(\cdot), \quad t > 0, \tag{20}$$

$$u''_{n, m}(t) + \lambda_{n, m}^2 \omega^2 u_{n, m}(t) = g_2(t) f_{2\,n, m}(\cdot), \quad t < 0, \tag{21}$$

where

$$\lambda_{n, m}^2 = \frac{\mu_n^2 + \mu_m^2}{1 + \mu_n^2 + \mu_m^2}, \quad \mu_n = \frac{n\pi}{l}, \quad \mu_m = \frac{m\pi}{l}, \quad n, m \in \mathbb{N}.$$

Taking (18) into account from the conditions (5) we derive:

$$\lim_{t \to +0} J_{0+}^{1-\gamma} u_{n, m}(t) = \frac{2}{l} \int_0^l \int_0^l \lim_{t \to +0} J_{0+}^{1-\gamma} U(t, x, y) \sin(\mu_n x) \cdot \sin(\mu_m y) \, dx \, dy$$

$$= \frac{2}{l} \int_0^l \int_0^l \lim_{t \to -0} U(t, x, y) \sin(\mu_n x) \sin(\mu_m y) \, dx \, dy = \lim_{t \to -0} u_{n, m}(t), \tag{22}$$

$$\lim_{t \to +0} J_{0+}^{1-\alpha} \frac{d}{dt} J_{0+}^{1-\gamma} u_{n, m}(t) = \frac{2}{l} \int_0^l \int_0^l \lim_{t \to +0} J_{0+}^{1-\alpha} \frac{d}{dt} J_{0+}^{1-\gamma} U(t, x, y) \sin(\mu_n x) \sin(\mu_m y) \, dx \, dy$$

$$= \frac{2}{l} \int\limits_0^l \int\limits_0^l \lim_{t \to -0} \frac{d}{dt} U(t, x, y) \sin(\mu_n x) \sin(\mu_m y) \, dx \, dy = \lim_{t \to -0} \frac{d}{dt} u_{n,m}(t). \tag{23}$$

Analogously we find from condition (4) that:

$$u_{n,m}(-a) = u_{n,m}(b) + \varphi_{n,m}, \tag{24}$$

where

$$\varphi_{n,m} = \frac{2}{l} \int\limits_0^l \int\limits_0^l \varphi(x, y) \sin(\mu_n x) \sin(\mu_m y) \, dx \, dy, \quad n, m = 1, 2, \dots$$

By applying Lemma 1, for (20) and (21) we obtain the general forms of solutions:

$$u_{n,m}(t) = A_{1n,m} t^{\gamma-1} E_{\alpha,\gamma}\left(-\lambda_{n,m}^2 t^\alpha\right) + f_{1n,m}(\cdot) h_{1n}(t), \quad t > 0, \tag{25}$$

$$u_{n,m}(t) = A_{2n,m} \sin \lambda_{n,m} \omega t + A_{3n,m} \cos \lambda_{n,m} \omega t + f_{2n,m}(\cdot) h_{2n,m}(t), \quad t < 0, \tag{26}$$

where $A_{in,m}$ are arbitrary constants, $i = \overline{1,3}, \quad n, m = 1, 2, \dots$

$$h_{1n,m}(t) = \int\limits_0^t (t-s)^{\alpha-1} E_{\alpha,\alpha}\left(-\lambda_{n,m}^2 (t-s)^\alpha\right) g_1(s) \, ds,$$

$$h_{2n,m}(t) = \frac{1}{\lambda_{n,m}\omega} \int\limits_0^t \sin(\lambda_{n,m}\omega(t-s)) g_2(s) \, ds.$$

Taking into account that $h_{1n,m}(0) = h_{2n,m}(0) = 0$ and satisfying functions (25) and (26) to conditions (22) and (23), we obtain the following systems of algebraic equations:

$$A_{2n,m} = -\frac{\lambda_{n,m}}{\omega} A_{1n,m}, \quad A_{3n,m} = A_{1n,m}. \tag{27}$$

Applying the condition (24) and representation (27) to (25) and (26), we derive:

$$A_{1n,m} = \frac{\varphi_{n,m} + f_{1n,m}(\cdot) h_{1n,m}(b) - f_{2n,m}(\cdot) h_{2n,m}(-a)}{\Delta_{n,m}(\omega)}, \tag{28}$$

if there holds the condition:

$$\Delta_{n,m}(\omega) = \lambda_{n,m}\omega^{-1} \sin(\lambda_{n,m}\omega a) + \cos(\lambda_{n,m}\omega a) - b^{\gamma-1} E_{\alpha,\gamma}\left(-\lambda_{n,m}^2 b^\alpha\right) \neq 0. \tag{29}$$

Substituting (28) into (27), for (25) and (26) we obtain the system of countable systems of nonlinear integral equations (SCSNIE):

$$u_{n,m}(t, \omega) = \mathbb{I}_1(t; u_{n,m})$$

$$\equiv \varphi_{n,m} \eta_{1n,m}(t, \omega) + f_{1n,m}(\cdot) \eta_{2n,m}(t, \omega) + f_{2n,m}(\cdot) \eta_{3n,m}(t, \omega), \quad t > 0, \tag{30}$$

$$u_{n,m}(t, \omega) = \mathbb{I}_2(t; u_{n,m})$$

$$\equiv \varphi_{n,m} \xi_{1n,m}(t, \omega) + f_{1n,m}(\cdot) \xi_{2n,m}(t, \omega) + f_{2n,m}(\cdot) \xi_{3n,m}(t, \omega), \quad t < 0, \tag{31}$$

where

$$f_{1\,n,m}(\cdot) = \int\limits_0^l \int\limits_0^l f_1\left(x,\, y,\, \int\limits_0^b \int\limits_0^l \int\limits_0^l \Theta_1\left(\theta,\, \zeta,\, \varsigma,\, \sum_{i,j=1}^{\infty} \theta^{1-\gamma} u_{i,j}(\theta)\, \vartheta_{i,j}(\zeta,\, \varsigma)\right) d\theta\, d\zeta\, d\varsigma\right) \vartheta_{n,m}(x,\, y)\, dx\, dy,$$

$$f_{2\,n,m}(\cdot) = \int\limits_0^l \int\limits_0^l f_2\left(x,\, y,\, \int\limits_{-a}^0 \int\limits_0^l \int\limits_0^l \Theta_2\left(\theta,\, \zeta,\, \varsigma,\, \sum_{i,j=1}^{\infty} u_{i,j}(\theta)\, \vartheta_{i,j}(\zeta,\, \varsigma)\right) d\theta\, d\zeta\, d\varsigma\right) \vartheta_{n,m}(x,\, y)\, dx\, dy,$$

$$\eta_{1\,n,m}(t,\, \omega) = \frac{t^{\gamma-1}}{\Delta_{n,m}(\omega)}\, E_{\alpha,\gamma}\left(-\lambda_{n,m}^2\, t^\alpha\right),\quad \eta_{2\,n,m}(t,\, \omega) = h_{1\,n,m}(t) + h_{1\,n,m}(b)\, \eta_{1\,n,m}(t,\, \omega),$$

$$\eta_{3\,n,m}(t,\, \omega) = -h_{2\,n,m}(-a)\, \eta_{1\,n,m}(t,\, \omega),\quad \xi_{1\,n,m}(t,\, \omega) = \frac{1}{\Delta_{n,m}(\omega)}\left[\sin(\lambda_{n,m}\omega t) + \cos(\lambda_{n,m}\omega t)\right],$$

$$\xi_{2\,n,m}(t,\, \omega) = h_{1\,n,m}(b)\, \xi_{1\,n,m}(t,\, \omega),\quad \xi_{3\,n,m}(t,\, \omega) = h_{2\,n,m}(t) + h_{2\,n,m}(-a)\, \xi_{1\,n,m}(t,\, \omega).$$

## 5. Solvability of SCSNIE (30) and (31)

Now we consider the case, when condition (29) is violated. Let $\Delta_{k,s}(\omega) = 0$ be for all $\omega$. Then the considering problem ($\varphi(x,\, y) \equiv 0$) has the nontrivial solution:

$$V_{k,s}(t,\, x,\, y) = v_{k,s}(t)\, \vartheta_{k,s}(x,\, y),\quad (t,\, x,\, y) \in \Omega, \tag{32}$$

where

$$v_{k,s}(t) = \begin{cases} t^{\gamma-1} E_{\alpha,\gamma}\left(-\lambda_{k,s}^2\, t^\alpha\right) + f_{1\,k,s}(\cdot)\, h_{1\,k,s}(t), & t > 0, \\ \sin\lambda_{k,s}\omega t + \cos\lambda_{k,s}\omega t + f_{2\,k,s}(\cdot)\, h_{2\,k,s}(t), & t < 0. \end{cases}$$

From $\Delta_{n,m}(\omega) = 0$ we come to the trigonometric equation:

$$\sqrt{1 + \frac{\lambda_{n,m}^2}{\omega^2}}\, \sin(\lambda_{n,m}\omega a + \rho_{n,m}) - b^{\gamma-1} E_{\alpha,\gamma}\left(-\lambda_{n,m}^2\, b^\alpha\right) = 0, \tag{33}$$

where $\rho_{n,m} = \arcsin\left(\dfrac{\omega}{\sqrt{\omega^2 + \lambda_{n,m}^2}}\right)$. From this we obtain that the quantity $\Delta_{n,m}(\omega)$ vanishes at the values:

$$\omega = \frac{1}{\lambda_{n,m}\, a}\left[(-1)^z \arcsin\frac{\omega\, b^{\gamma-1} E_{\alpha,\gamma}\left(-\lambda_{n,m}^2\, b^\alpha\right)}{\sqrt{\omega^2 + \lambda_{n,m}^2}} + \pi z - \rho_{n,m}\right],\quad z \in \mathbb{N}.$$

The set of positive solutions $\Im$ of trigonometric Equation (33) with respect to spectral parameter $\omega$ is called a set of irregular values of the spectral parameter $\omega$. The set of the remaining values of the spectral parameter $\aleph = (0;\, \infty) \setminus \Im$ is called a set of regular values of the spectral parameter $\omega$. For all regular values of the spectral parameter $\omega$, the quantity $\Delta_{n,m}(\omega)$ is nonzero. So, for large $n$, $m$ the values of $\Delta_{n,m}(\omega)$ can not become quite small and there the problem of "small denominators" does not arise. Therefore, for regular values of the spectral parameter $\omega$, the quantity $\Delta_{n,m}(\omega)$ is separated from zero.

Indeed, from the relations:

$$\lambda_{n,m}^2 = \frac{\mu_n^2 + \mu_m^2}{1 + \mu_n^2 + \mu_m^2},\quad \mu_n = \frac{n\pi}{l},\quad \mu_m = \frac{m\pi}{l},\quad n,\, m \in \mathbb{N}$$

we see that $\lambda_{n,m}^2 \to 1$ as $n, m \to \infty$. Therefore, for regular values of the spectral parameter $\omega$ we have:

$$\lim_{n,m\to\infty} \Delta_{n,m}(\omega) = \frac{1}{\omega} \sin \omega a + \cos \omega a - b^{\gamma-1} E_{\alpha,\gamma}(-b^\alpha) \neq 0.$$

**Lemma 2.** *Suppose that $\gamma \in (0,1]$, $a, b$ are arbitrary positive real numbers. Then for regular values of the spectral parameter $\omega \in \aleph$ and for arbitrary $n$, $m$ there exists a positive constant $M_0$ such that there holds the following estimate:*

$$|\Delta_{n,m}(\omega)| \geq M_0 > 0. \tag{34}$$

**Proof.** From (33) for all $n$, $m$ and $a$, $b > 0$ we derive:

$$|\Delta_{n,m}(\omega, \nu)| \geq \left| \pm\sqrt{1 + \frac{\lambda_{n,m}^2(\nu)}{\omega^2}} - b^{\gamma-1} E_{\alpha,\gamma}\left(-\lambda_{n,m}^2 b^\alpha\right) \right|$$

$$\geq \left| 1 - b^{\gamma-1} E_{\alpha,\gamma}\left(-\lambda_{n,m}^2 b^\alpha\right) \right|.$$

We use the following properties of the Mittag–Leffler function (Volume 1, pp. 269–295) in [1]:

(1)  For all $k > 0$, $\alpha, \gamma \in (0;1]$, $\alpha \leq \gamma$, $t \geq 0$ the function $t^{\gamma-1} E_{\alpha,\gamma}(-k t^\alpha)$ is completely monotonous and there holds:

$$(-1)^s \left[ t^{\gamma-1} E_{\alpha,\gamma}(-k t^\alpha) \right]^{(s)} \geq 0, \quad s = 0, 1, 2, \ldots \tag{35}$$

(2)  For all $\alpha \in (0;2)$, $\gamma \in \mathbb{R}$ and $\arg z = \pi$ there takes place the following estimate:

$$|E_{\alpha,\gamma}(z)| \leq \frac{M_1}{1 + |z|}, \tag{36}$$

where $0 < M_1 = $ const does not depend from $z$.

Then, from the inequalities (35) and (36) we derive that there exists a number $M_0$ such that:

$$\left| 1 - b^{\gamma-1} E_{\alpha,\gamma}\left(-\lambda_{n,m}^2 b^\alpha\right) \right| = M_0 > 0.$$

Consequently, for regular values of the spectral parameter $\omega$ there takes place (34): $|\Delta_{n,m}(\omega)| \geq M_0 > 0$. Lemma 2 is proved.  $\square$

**Condition A.** Let the following be fulfilled:

$$\varphi(x, y) \in C^3[0; l]^2, \varphi_{xxxx}(x, y) \in L_2[0; l]^2, \varphi_{yyyy}(x, y) \in L_2[0; l]^2.$$

Then by integrating in parts four times over the variable $x$ the integral:

$$\varphi_{n,m} = \int_0^l \int_0^l \varphi(x, y) \vartheta_{n,m}(x, y) \, dx \, dy,$$

we derive that:

$$\varphi_{n,m} = \left(\frac{l}{\pi}\right)^4 \frac{\varphi_{n,m}^{(IV)}}{n^4}, \tag{37}$$

where,

$$\varphi_{n,m}^{(IV)} = \int_0^l \int_0^l \varphi_{xxxx}(x,y)\,\vartheta_{n,m}(x,y)\,dx\,dy, \tag{38}$$

$$\vartheta_{n,m}(x,y) = \frac{2}{l} \sin\frac{\pi n}{l} x \, \sin\frac{\pi m}{l} y.$$

Similarly, by integrating the integral (38) in parts four times with respect to the variable $y$ yields:

$$\varphi_{n,m}^{(IV)} = \left(\frac{l}{\pi}\right)^4 \frac{\varphi_{n,m}^{(VIII)}}{m^4}, \tag{39}$$

where

$$\varphi_{n,m}^{(VIII)} = \int_0^l \int_0^l \varphi_{xxxxyyyy}(x,y)\,\vartheta_{n,m}(x,y)\,dx\,dy. \tag{40}$$

Substituting (39) into (37), we obtain:

$$\varphi_{n,m} = \left(\frac{l}{\pi}\right)^8 \frac{\varphi_{n,m}^{(VIII)}}{n^4 m^4}. \tag{41}$$

Applying the Bessel inequality for the integral (40), we obtain the estimate:

$$\sum_{n,m=1}^{\infty} \left[\varphi_{n,m}^{(VIII)}\right]^2 = \sum_{n,m=1}^{\infty} \left[\int_0^l \int_0^l \varphi_{xxxxyyyy}(x,y)\,\vartheta_{n,m}(x,y)\,dx\,dy\right]^2$$

$$\leq \int_0^l \int_0^l \left[\varphi_{xxxxyyyy}(x,y)\right]^2 dx\,dy < \infty. \tag{42}$$

**Condition B.** Let the following be fulfilled:

$$f_i(x,y,u) \in C_{x,y,u}^{3,3,0}\left([0;l]^2 \times \mathbb{R}\right),\ f_{ixxxx}(x,y,u) \in L_2\left([0;l]^2 \times \mathbb{R}\right),$$

$$f_{iyyyy}(x,y,u) \in L_2\left([0;l]^2 \times \mathbb{R}\right),\ i=1,2,$$

where

$$L_2\left([0;l]^2 \times \mathbb{R}\right) = \left\{ f(x,y,u): \sqrt{\int_0^l \int_0^l |f(x,y,u)|^2 dx\,dy} < \infty \right\}.$$

Similarly to the case of condition **A**, we obtain:

$$f_{in,m}(\cdot) = \left(\frac{l}{\pi}\right)^8 \frac{f_{in,m}^{(VIII)}(\cdot)}{n^4 m^4}, \tag{43}$$

$$\sum_{n,m=1}^{\infty} \left[f_{in,m}^{(VIII)}(\cdot)\right]^2 \leq \int_0^l \int_0^l \left[f_{ixxxxyyyy}(x,y,\cdot)\right]^2 dx\,dy < \infty, \tag{44}$$

where

$$f_{in,m}^{(VIII)}(\cdot) = \frac{2}{l} \int_0^l \int_0^l f_{ixxxxyyyy}(x,y,\cdot) \sin\frac{\pi n}{l} x \, \sin\frac{\pi m}{l} y\,dx\,dy.$$

For all regular values of the spectral parameter $\omega \in \aleph$ the SCSNIE (30) and (31) are true. In order to prove the unique solvability of SCSNIE (30) and (31), we introduce the following well-knowing spaces.

Space $B_2[-a; b]$ of sequences of continuous functions $\{u_{n,m}(t)\}_{n,m=1}^{\infty}$ on the segment $[-a; b]$ with the norm:

$$\| u(t) \|_{B_2[-a; b]} = \| u(t) \|_{B_2[-a; 0]} + \| u(t) \|_{B_2[0; b]}$$

$$= \sqrt{\sum_{n,m=1}^{\infty} \left( \max_{t \in [-a; 0]} | u_{n,m}(t) | \right)^2} + \sqrt{\sum_{n,m=1}^{\infty} \left( \max_{t \in [0; b]} | u_{n,m}(t) | \right)^2} < \infty.$$

The space $L_2[0; l]^2$ of square-summable functions on the domain $[0; l]^2 = [0; l] \times [0; l]$ with the norm:

$$\| \vartheta(x, y) \|_{L_2[0; l]^2} = \sqrt{\int_0^l \int_0^l | \vartheta(x, y) |^2 \, dx \, dy} < \infty.$$

On the basis of lemma 2, Conditions **A** and **B** for regular spectral values from the sets $\aleph$ we prove that it holds.

**Theorem 1.** *Suppose that the following conditions and Conditions **A**, **B** are fulfilled:*

(1) $\chi_{11} = \max\limits_{i=\overline{1,3}} \max\limits_{n,m \in \mathbb{N}} \max\limits_{t \in [0; b]} | t^{1-\gamma} \eta_{inm}(t, \omega) | < \infty;$  $\chi_{21} = \max\limits_{i=\overline{1,3}} \max\limits_{n,m \in \mathbb{N}} \max\limits_{t \in [-a; 0]} | \xi_{inm}(t, \omega) | < \infty;$

(2) $\chi_{30} = \| \varphi_{xxxxyyyy}(x, y) \|_{L_2[0; l]^2} < \infty;$  $\chi_{3i} = \| f_{i\,xxxxyyyy}(x, y, \gamma) \|_{L_2[0; l]^2} < \infty;$

(3) $| f_{i\,xxxxyyyy}(x, y, \gamma_1) - f_{i\,xxxxyyyy}(x, y, \gamma_2) | \le K_i(x, y) | \gamma_1 - \gamma_2 |,$

$K_{0i} = \| K_i(x, y) \|_{L_2[0; l]^2} < \infty;$

(4) $| \Theta_i(\xi, x, y, u_1) - \Theta_i(\xi, x, y, u_2) | \le \Theta_{1i}(x, y) | u_1 - u_2 |,$

$\Theta_{2i} = \| \Theta_{1i}(x, y) \|_{L_2[0; l]^2} < \infty,$  $i = 1, 2;$

(5) $\rho = \gamma_2(\gamma_1 + \gamma_3)\gamma_4 < 1,$  $\gamma_4 = \max\{b K_{01}\Theta_{21}; a K_{02}\Theta_{22}\}.$

*Then SCSNIE (30) and (31) are uniquely solvable in the spaces $B_2[-a; 0]$ and $B_2[0; b]$, respectively for all regular values of the spectral parameter $\omega \in \aleph$.*

**Proof.** We use the method of compressing mappings in the Banach spaces $B_2[-a; 0]$ and $B_2[0; b]$. Successive approximations are defined as follows:

$$\begin{cases} u_{n,m}^0(t, \omega) = \varphi_{n,m}\eta_{1\,n,m}(t, \omega), \ u_{n,m}^{k+1} = \mathbb{I}_1(t; u_{n,m}^k), \ k = 0, 1, 2, \ldots, \ t > 0, \\ u_{n,m}^0(t, \omega) = \varphi_{n,m}\xi_{1\,n,m}(t, \omega), \ u_{n,m}^{k+1} = \mathbb{I}_2(t; u_{n,m}^k), \ t < 0, \ \omega \in \aleph. \end{cases} \quad (45)$$

When $t > 0$, by virtue of the first condition of the theorem and applying Cauchy–Schwarz inequality and properties (41) and (42) to the approximations (45) for the zero approximation $u_{n,m}^0(t, \omega)$ with the norm in $B_2[0; b]$ obtains the estimate:

$$\left\| t^{1-\gamma} u^0(t, \omega) \right\|_{B_2[0; b]} \le \max_{t \in [0; b]} \sum_{n,m=1}^{\infty} | \varphi_n | \left| t^{1-\gamma} \eta_{1\,nm}(t, \omega) \right|$$

$$\le \max_{n,m \in \mathbb{N}} \max_{t \in [0; b]} \left| t^{1-\gamma} \eta_{1\,nm}(t, \omega) \right| \sum_{n,m=1}^{\infty} \left| \left( \frac{l}{\pi} \right)^8 \frac{\varphi_{n,m}^{(VIII)}}{n^4 m^4} \right|$$

$$\le \chi_{11} \left( \frac{l}{\pi} \right)^8 \sum_{n,m=1}^{\infty} \left| \frac{1}{n^4 m^4} \right| \cdot \left| \varphi_{n,m}^{(VIII)} \right| \le \gamma_1 \sqrt{\sum_{n,m=1}^{\infty} \frac{1}{n^8 m^8}} \sqrt{\sum_{n,m=1}^{\infty} \left| \varphi_{n,m}^{(VIII)} \right|^2}$$

$$
\leq \gamma_1 \gamma_2 \sqrt{\int_0^l \int_0^l \left[ \varphi_{xxxxyyyy}(x, y) \right]^2 dx \, dy} = \gamma_1 \gamma_2 \gamma_{30} < \infty, \tag{46}
$$

where $\gamma_1 = \chi_{11} \left( \dfrac{l}{\pi} \right)^8$, $\gamma_2 = \sqrt{\displaystyle\sum_{n,m=1}^{\infty} \frac{1}{n^8 m^8}}$.

Similarly, by virtue of the conditions of the theorem and applying Cauchy–Schwarz inequality and properties (43) and (44) for the first difference of approximations (45), we derive:

$$
\left\| t^{1-\gamma} \left( u^1(t, \omega) - u^0(t, \omega) \right) \right\|_{B_2[0;b]} \leq \max_{t \in [0;b]} \sum_{n,m=1}^{\infty} \left| f^0_{1n,m}(\cdot) \right| \cdot \left| t^{1-\gamma} \eta_{2nm}(t, \omega) \right|
$$

$$
+ \max_{t \in [0;b]} \sum_{n,m=1}^{\infty} \left| f^0_{2n,m}(\cdot) \right| \cdot \left| t^{1-\gamma} \eta_{3nm}(t, \omega) \right|
$$

$$
\leq \max_{n,m \in \mathbb{N}} \max_{t \in [0;b]} \left| t^{1-\gamma} \eta_{2nm}(t, \omega) \right| \left( \frac{l}{\pi} \right)^8 \sum_{n,m=1}^{\infty} \left| \frac{f^{(VIII)}_{1n,m}(\cdot)}{n^4 m^4} \right|
$$

$$
+ \max_{n,m \in \mathbb{N}} \max_{t \in [0;b]} \left| t^{1-\gamma} \eta_{3nm}(t, \omega) \right| \left( \frac{l}{\pi} \right)^8 \sum_{n,m=1}^{\infty} \left| \frac{f^{(VIII)}_{2n,m}(\cdot)}{n^4 m^4} \right|
$$

$$
\leq \chi_{11} \left( \frac{l}{\pi} \right)^8 \left[ \sum_{n,m=1}^{\infty} \frac{1}{n^4 m^4} \left| f^{(VIII)}_{1n,m}(\cdot) \right| + \sum_{n,m=1}^{\infty} \frac{1}{n^4 m^4} \left| f^{(VIII)}_{2n,m}(\cdot) \right| \right]
$$

$$
\leq \gamma_1 \sqrt{\sum_{n,m=1}^{\infty} \frac{1}{n^8 m^8}} \left[ \sqrt{\sum_{n,m=1}^{\infty} \left| f^{(VIII)}_{1n,m}(\cdot) \right|^2} + \sqrt{\sum_{n,m=1}^{\infty} \left| f^{(VIII)}_{2n,m}(\cdot) \right|^2} \right]
$$

$$
\leq \gamma_1 \gamma_2 \left[ \sqrt{\int_0^l \int_0^l \left[ f_{1xxxxyyyy}(x, y, \cdot) \right]^2 dx \, dy} \right.
$$

$$
\left. + \sqrt{\int_0^l \int_0^l \left[ f_{2xxxxyyyy}(x, y, \cdot) \right]^2 dx \, dy} \right] = \gamma_1 \gamma_2 (\chi_{31} + \chi_{32}) < \infty, \tag{47}
$$

where

$$
f^k_{1n,m}(\cdot) = \int_0^l \int_0^l f_1 \left( x, y, \int_0^b \int_0^l \int_0^l \Theta_1 \left( \theta, \zeta, \varsigma, \sum_{i,j=1}^{\infty} \theta^{1-\gamma} u^k_{i,j}(\theta) \vartheta_{i,j}(\zeta, \varsigma) \right) d\theta \, d\zeta \, d\varsigma \right) \vartheta_{n,m}(x, y) \, dx \, dy,
$$

$$
f^k_{2n,m}(\cdot) = \int_0^l \int_0^l f_2 \left( x, y, \int_{-a}^0 \int_0^l \int_0^l \Theta_2 \left( \theta, \zeta, \varsigma, \sum_{i,j=1}^{\infty} u^k_{i,j}(\theta) \vartheta_{i,j}(\zeta, \varsigma) \right) d\theta \, d\zeta \, d\varsigma \right) \vartheta_{n,m}(x, y) \, dx \, dy,
$$

$k = 0, 1, 2, \ldots$

We use the conditions of theorem, Cauchy–Schwarz inequality, and Bessel inequality for the arbitrary difference $u^{k+1}_{n,m}(t, \omega) - u^k_{n,m}(t, \omega)$ with the norm in $B_2[0; b]$. Then we derive from (45) the following estimate:

$$
\left\| t^{1-\gamma} \left( u^{k+1}(t, \omega) - u^k(t, \omega) \right) \right\|_{B_2[0;b]}
$$

$$
\leq \max_{t \in [0;b]} \sum_{n,m=1}^{\infty} \left| f_{1n,m}^{k}(\cdot) - f_{1n,m}^{k-1}(\cdot) \right| \left| t^{1-\gamma} \eta_{2nm}(t,\omega) \right|
$$

$$
+ \max_{t \in [0;b]} \sum_{n,m=1}^{\infty} \left| f_{2n,m}^{k}(\cdot) - f_{2n,m}^{k-1}(\cdot) \right| \cdot \left| t^{1-\gamma} \eta_{3nm}(t,\omega) \right|
$$

$$
\leq \gamma_1 \left[ \sum_{n,m=1}^{\infty} \frac{1}{n^4 m^4} \left| f_{1n,m}^{k\,(VIII)}(\cdot) - f_{1n,m}^{k-1\,(VIII)}(\cdot) \right| + \sum_{n,m=1}^{\infty} \frac{1}{n^4 m^4} \left| f_{2n,m}^{k\,(VIII)}(\cdot) - f_{2n,m}^{k-1\,(VIII)}(\cdot) \right| \right]
$$

$$
\leq \gamma_1 \sqrt{\sum_{n,m=1}^{\infty} \frac{1}{n^8 m^8}} \left[ \sqrt{\int_0^l \int_0^l \left| f_{1xxxxyyyy}^{k}(x,y,\cdot) - f_{1xxxxyyyy}^{k-1}(x,y,\cdot) \right|^2 dx\,dy} \right.
$$

$$
\left. + \sqrt{\int_0^l \int_0^l \left| f_{2xxxxyyyy}^{k}(x,y,\cdot) - f_{2xxxxyyyy}^{k-1}(x,y,\cdot) \right|^2 dx\,dy} \right]
$$

$$
\leq \gamma_1 \gamma_2 \left[ \sqrt{\int_0^l \int_0^l |K_1(x,y)|^2 dx\,dy} \int_0^b \int_0^l \int_0^l \left| \Theta_1^{k}(\cdot) - \Theta_1^{k-1}(\cdot) \right| d\theta\,d\zeta\,d\varsigma \right.
$$

$$
\left. + \sqrt{\int_0^l \int_0^l |K_2(x,y)|^2 dx\,dy} \int_{-a}^0 \int_0^l \int_0^l \left| \Theta_2^{k}(\cdot) - \Theta_2^{k-1}(\cdot) \right| d\theta\,d\zeta\,d\varsigma \right]
$$

$$
\leq \gamma_1 \gamma_2 \left[ K_{01} \int_0^b \int_0^l \int_0^l |\Theta_{11}(\zeta,\varsigma)| \cdot \left| \sum_{i,j=1}^{\infty} \theta^{1-\gamma} \left[ u_{i,j}^{k}(\theta) - u_{i,j}^{k-1}(\theta) \right] \vartheta_{i,j}(\zeta,\varsigma) \right| d\theta\,d\zeta\,d\varsigma \right.
$$

$$
\left. + K_{02} \int_{-a}^0 \int_0^l \int_0^l |\Theta_{12}(\zeta,\varsigma)| \cdot \left| \sum_{i,j=1}^{\infty} \left[ u_{i,j}^{k}(\theta) - u_{i,j}^{k-1}(\theta) \right] \vartheta_{i,j}(\zeta,\varsigma) \right| d\theta\,d\zeta\,d\varsigma \right]
$$

$$
\leq \gamma_1 \gamma_2 \left[ K_{01} \| \Theta_{11}(x,y) \|_{L_2[0;l]^2} \int_0^b \left\| \theta^{1-\gamma} \left( u^{k}(\theta,\omega) - u^{k-1}(\theta,\omega) \right) \right\|_{B_2[0;b]} d\theta \right.
$$

$$
\left. + K_{02} \| \Theta_{12}(x,y) \|_{L_2[0;l]^2} \int_{-a}^0 \left\| u^{k}(\theta,\omega) - u^{k-1}(\theta,\omega) \right\|_{B_2[-a;0]} d\theta \right]
$$

$$
\leq \gamma_1 \gamma_2 \left[ b K_{01} \Theta_{21} \left\| t^{1-\gamma} \left( u^{k}(t,\omega) - u^{k-1}(t,\omega) \right) \right\|_{B_2[0;b]} \right.
$$

$$
\left. + a K_{02} \Theta_{22} \left\| u^{k}(t,\omega) - u^{k-1}(t,\omega) \right\|_{B_2[-a;0]} \right]. \tag{48}
$$

When $t < 0$, by virtue of the conditions of the theorem and applying the Cauchy–Schwarz inequality and Bessel inequality to (45) we similarly obtain the following estimates:

$$
\left\| u^{0}(t,\omega) \right\|_{B_2[-a;0]} \leq \max_{t \in [-a;0]} \sum_{n,m=1}^{\infty} |\varphi_n| \, |\xi_{1nm}(t,\omega)|
$$

$$
\leq \gamma_2 \gamma_3 \left\| \varphi_{xxxxyyyy}(x,y) \right\|_{L_2[0;l]^2} < \infty, \tag{49}
$$

where $\gamma_3 = \chi_{21} \left( \frac{l}{\pi} \right)^8$;

$$\left\| u^1(t, \omega) - u^0(t, \omega) \right\|_{B_2[-a;0]} \leq \max_{t \in [-a;0]} \sum_{n,m=1}^{\infty} \left| f^0_{1n,m}(\cdot) \right| \cdot \left| \xi_{2nm}(t, \omega) \right|$$

$$+ \max_{t \in [-a;0]} \sum_{n,m=1}^{\infty} \left| f^0_{2n,m}(\cdot) \right| \cdot \left| \xi_{3nm}(t, \omega) \right| \leq \gamma_2 \gamma_3 \left[ \left\| f_{1xxxxyyyy}(x, y, \cdot) \right\|_{L_2[0;l]^2} \right.$$

$$\left. + \left\| f_{2xxxxyyyy}(x, y, \cdot) \right\|_{L_2[0;l]^2} \right] = \gamma_2 \gamma_3 \left( \chi_{31} + \chi_{32} \right) < \infty; \tag{50}$$

$$\left\| u^{k+1}(t, \omega) - u^k(t, \omega) \right\|_{B_2[-a;0]} \leq \max_{t \in [-a;0]} \sum_{n,m=1}^{\infty} \left| f^k_{1n,m}(\cdot) - f^{k-1}_{1n,m}(\cdot) \right| \left| \xi_{2nm}(t, \omega) \right|$$

$$+ \max_{t \in [-a;0]} \sum_{n,m=1}^{\infty} \left| f^k_{2n,m}(\cdot) - f^{k-1}_{2n,m}(\cdot) \right| \cdot \left| \xi_{3nm}(t, \omega) \right|$$

$$\leq \gamma_2 \gamma_3 \left[ \left\| K_1(x, y) \right\|_{L_2[0;l]^2} \int_0^b \int_0^l \int_0^l \left| \Theta^k_1(\cdot) - \Theta^{k-1}_1(\cdot) \right| d\theta \, d\zeta \, d\varsigma \right.$$

$$\left. + \left\| K_2(x, y) \right\|_{L_2[0;l]^2} \int_{-a}^0 \int_0^l \int_0^l \left| \Theta^k_2(\cdot) - \Theta^{k-1}_2(\cdot) \right| d\theta \, d\zeta \, d\varsigma \right]$$

$$\leq \gamma_2 \gamma_3 \left[ K_{01} \left\| \Theta_{11}(x, y) \right\|_{L_2[0;l]^2} \int_0^b \left\| \theta^{1-\gamma} \left( u^k(\theta, \omega) - u^{k-1}(\theta, \omega) \right) \right\|_{B_2[0;b]} d\theta \right.$$

$$\left. + K_{02} \left\| \Theta_{12}(x, y) \right\|_{L_2[0;l]^2} \int_{-a}^0 \left\| u^k(\theta, \omega) - u^{k-1}(\theta, \omega) \right\|_{B_2[-a;0]} d\theta \right]$$

$$\leq \gamma_2 \gamma_3 \left[ b \, K_{01} \Theta_{21} \left\| t^{1-\gamma} \left( u^k(t, \omega) - u^{k-1}(t, \omega) \right) \right\|_{B_2[0;b]} \right.$$

$$\left. + a \, K_{02} \Theta_{22} \left\| u^k(t, \omega) - u^{k-1}(t, \omega) \right\|_{B_2[-a;0]} \right]. \tag{51}$$

Adding inequalities (48) and (51), we obtain:

$$\left\| u^{k+1}(t, \omega) - u^k(t, \omega) \right\|_{B_2[-a;b]} \leq \rho \left\| u^k(t, \omega) - u^{k-1}(t, \omega) \right\|_{B_2[-a;b]}, \tag{52}$$

where $\rho = \gamma_2 (\gamma_1 + \gamma_3) \gamma_4$, $\gamma_4 = \max \{ b \, K_{01} \Theta_{21}; a \, K_{02} \Theta_{22} \}$.

According to the last condition of the theorem there is $\rho = \gamma_2 (\gamma_1 + \gamma_3) \gamma_4 < 1$. Therefore from the estimates (46), (47), (49), (50) and (52) implies that the operators on the right side of (30), and (31) are compressive and there exists a unique fixed point for these operators. Therefore the SCSNIE (30) and (31) are uniquely solvable in the space $B_2[-a; b]$ for regular spectral values of parameter $\omega \in \aleph$. Theorem 1 is thus proved. □

## 6. Convergence of Fourier Series

Substituting SCSNIE (30) and (31) into the Fourier series (17), we obtain:

$$U(t, x, y, \omega) = \sum_{n,m=1}^{\infty} \vartheta_{n,m}(x, y) \left[ \varphi_{n,m} \eta_{1n,m}(t, \omega) + \eta_{2n,m}(t, \omega) f_{1n,m}(\cdot) \right.$$

$$+\eta_{3\,n,\,m}(t,\omega)\,f_{2\,n,\,m}(\cdot)],\ (t,x,y)\in\Omega_1, \tag{53}$$

$$U(t,x,y,\omega)=\sum_{n,\,m=1}^{\infty}\vartheta_{n,\,m}(x,y)\,[\varphi_{n,\,m}\,\xi_{1\,n,\,m}(t,\omega)+\xi_{2\,n,\,m}(t,\omega)\,f_{1\,n,\,m}(\cdot)$$

$$+\xi_{3\,n,\,m}(t,\omega)\,f_{2\,n,\,m}(\cdot)],\ (t,x,y)\in\Omega_2, \tag{54}$$

where

$$f_{1\,n,\,m}(\cdot)=\int_0^l\int_0^l f_1\left(x,y,\int_0^b\int_0^l\int_0^l\Theta_1\left(\theta,\zeta,\varsigma,\sum_{i,\,j=1}^{\infty}\theta^{1-\gamma}u_{i,\,j}(\theta)\,\vartheta_{i,\,j}(\zeta,\varsigma)\right)d\theta\,d\zeta\,d\varsigma\right)\vartheta_{n,\,m}(x,y)\,dx\,dy,$$

$$f_{2\,n,\,m}(\cdot)=\int_0^l\int_0^l f_2\left(x,y,\int_{-a}^0\int_0^l\int_0^l\Theta_2\left(\theta,\zeta,\varsigma,\sum_{i,\,j=1}^{\infty}u_{i,\,j}(\theta)\,\vartheta_{i,\,j}(\zeta,\varsigma)\right)d\theta\,d\zeta\,d\varsigma\right)\vartheta_{n,\,m}(x,y)\,dx\,dy.$$

**Theorem 2.** *Let conditions of the Theorem 1 be fulfilled. Then for regular values of the spectral parameter $\omega\in\aleph$ the Fourier series (53) and (54) are convergent absolutely and uniformly in the domain $\Omega_1$ and $\Omega_2$, respectively. The series (53) and (54) possess the Properties (2).*

**Proof.** We prove the absolutely and uniformly convergence of series (53) and (54). Similarly to the estimates (46), (47) and (49), (50), we obtain:

$$\left|t^{1-\gamma}U(t,x,y,\omega)\right|\le\max_{t\in[0;b]}\sum_{n,\,m=1}^{\infty}\left|t^{1-\gamma}u_{n,\,m}(t,\omega)\right|\cdot|\vartheta_{n\,m}(x,y)|$$

$$\le\frac{2}{l}\max_{t\in[0;b]}\left[\sum_{n,\,m=1}^{\infty}|\varphi_{n,\,m}(\cdot)|\cdot\left|t^{1-\gamma}\eta_{1\,n\,m}(t,\omega)\right|+\sum_{n,\,m=1}^{\infty}|f_{1\,n,\,m}(\cdot)|\cdot\left|t^{1-\gamma}\eta_{2\,n\,m}(t,\omega)\right|\right.$$

$$\left.+\sum_{n,\,m=1}^{\infty}|f_{2\,n,\,m}(\cdot)|\cdot\left|t^{1-\gamma}\eta_{3\,n\,m}(t,\omega)\right|\right]$$

$$\le\frac{2}{l}\gamma_1\left[\sum_{n,\,m=1}^{\infty}\frac{1}{n^4m^4}\left|\varphi_{n,\,m}^{(VIII)}\right|+\sum_{n,\,m=1}^{\infty}\frac{1}{n^4m^4}\left|f_{1\,n,\,m}^{(VIII)}(\cdot)\right|+\sum_{n,\,m=1}^{\infty}\frac{1}{n^4m^4}\left|f_{2\,n,\,m}^{(VIII)}(\cdot)\right|\right]$$

$$\le\gamma_5\left[\left\|\varphi_{xxxxyyyy}(x,y)\right\|_{L_2[0;l]^2}+\left\|f_{1xxxxyyyy}(x,y,\cdot)\right\|_{L_2[0;l]^2}\right.$$

$$\left.+\left\|f_{2xxxxyyyy}(x,y,\cdot)\right\|_{L_2[0;l]^2}\right]=\gamma_5(\chi_{30}+\chi_{31}+\chi_{32})<\infty,\ \gamma_5=\frac{2}{l}\gamma_1\gamma_2; \tag{55}$$

$$|U(t,x,y,\omega)|\le\max_{t\in[-a;0]}\sum_{n,\,m=1}^{\infty}|u_{n,\,m}(t,\omega)|\cdot|\vartheta_{n\,m}(x,y)|$$

$$\le\frac{2}{l}\max_{t\in[-a;0]}\left[\sum_{n,\,m=1}^{\infty}|\varphi_{n,\,m}(\cdot)|\cdot|\xi_{1\,n\,m}(t,\omega)|+\sum_{n,\,m=1}^{\infty}|f_{1\,n,\,m}(\cdot)|\cdot|\xi_{2\,n\,m}(t,\omega)|\right.$$

$$\left.+\sum_{n,\,m=1}^{\infty}|f_{2\,n,\,m}(\cdot)|\cdot|\xi_{3\,n\,m}(t,\omega)|\right]\le\frac{2}{l}\gamma_3\left[\sum_{n,\,m=1}^{\infty}\frac{1}{n^4m^4}\left|\varphi_{n,\,m}^{(VIII)}\right|\right.$$

$$\left.+\sum_{n,\,m=1}^{\infty}\frac{1}{n^4m^4}\left|f_{1\,n,\,m}^{(VIII)}(\cdot)\right|+\sum_{n,\,m=1}^{\infty}\frac{1}{n^4m^4}\left|f_{2\,n,\,m}^{(VIII)}(\cdot)\right|\right]$$

$$\le\gamma_6(\chi_{30}+\chi_{31}+\chi_{32})<\infty,\ \gamma_6=\frac{2}{l}\gamma_2\gamma_3. \tag{56}$$

Similarly to case of (55) and (56), it is easy to prove that the following series are convergent absolutely and uniformly in the domain $\Omega_1$ and $\Omega_2$, respectively:

$$t^{1-\gamma} D^{\alpha,\gamma} U(t,x,y,\omega) = \sum_{n,m=1}^{\infty} t^{1-\gamma} D^{\alpha,\gamma} u_{n,m}(t,\omega)\,\vartheta_{n,m}(x,y), \quad (t,x,y) \in \Omega_1, \tag{57}$$

$$t^{1-\gamma}\frac{\partial^k U(t,x,y,\omega)}{\partial x^k} = (-1)^{k+1} \sum_{n,m=1}^{\infty} t^{1-\gamma} u_{n,m}(t,\omega)\,\mu_n^k \vartheta_{n,m}(x,y),\ k=1,2,\ (t,x,y)\in\Omega_1, \tag{58}$$

$$t^{1-\gamma}\frac{\partial^k U(t,x,y,\omega)}{\partial y^k} = (-1)^{k+1} \sum_{n,m=1}^{\infty} t^{1-\gamma} u_{n,m}(t,\omega)\,\mu_m^k \vartheta_{n,m}(x,y),\ k=1,2,\ (t,x,y)\in\Omega_1, \tag{59}$$

$$\frac{\partial^2 U(t,x,y,\omega)}{\partial t^2} = \sum_{n,m=1}^{\infty} \frac{d^2 u_{n,m}(t,\omega)}{dt^2}\,\vartheta_{n,m}(x,y), \quad (t,x,y)\in\Omega_2, \tag{60}$$

$$\frac{\partial^k U(t,x,y,\omega)}{\partial x^k} = (-1)^{k+1} \sum_{n,m=1}^{\infty} u_{n,m}(t,\omega)\,\mu_n^k \vartheta_{n,m}(x,y),\ k=1,2,\ (t,x,y)\in\Omega_2, \tag{61}$$

$$\frac{\partial^k U(t,x,y,\omega)}{\partial y^k} = (-1)^{k+1} \sum_{n,m=1}^{\infty} u_{n,m}(t,\omega)\,\mu_m^k \vartheta_{n,m}(x,y),\ k=1,2,\ (t,x,y)\in\Omega_2. \tag{62}$$

Theorem 2 is proved. ☐

## 7. Irregular Value of Spectral Parameter $\omega$

We note that $\Delta_{n,m}(\omega) = 0$ for irregular values of the spectral parameter $\omega \in \Im$ and $n, m = k, s$ ($\gamma \neq 1$). Then, for the solvability of systems (25) and (26), it is necessary and sufficient that the orthogonality conditions are satisfied:

$$\varphi_{k,s} = \int_0^l \int_0^l \varphi(x,y)\,\vartheta_{k,s}(x,y)\,dx\,dy = 0. \tag{63}$$

In this case, by virtue of (32), the solutions of nonlocal problem are represented as:

$$U(t,x,y) = \sum_{k,s=1}^{\infty} C_{k,s}\left[t^{\gamma-1} E_{\alpha,\gamma}\left(-\lambda_{k,s}^2\, t^\alpha\right) + f_{1k,s}(\cdot)\,h_{1k,s}(t)\right]\vartheta_{k,s}(x,y), \quad (t,x,y)\in\Omega_1, \tag{64}$$

$$U(t,x,y) = \sum_{k,s=1}^{\infty} C_{k,s}\left[\sin\lambda_{k,s}\,\omega t + \cos\lambda_{k,s}\,\omega t + f_{2k,s}(\cdot)\,h_{2k,s}(t)\right]\vartheta_{k,s}(x,y),\ (t,x,y)\in\Omega_2, \tag{65}$$

where $k, s = k_1, ..., k_s$, $C_{k,s}$ are arbitrary constants.

The absolute and uniform convergence of the obtained series (64) and (65) is clear, since $C_{k,s}$ are arbitrary numbers. Them we can select that these series converge. We recall that the Fourier coefficient functions $f_{1k,s}(\cdot)$ and $f_{2k,s}(\cdot)$ in (64) and (65) satisfy the properties (43) and (44).

## 8. Conclusions

In this paper, we considered a nonlocal boundary value problem $T_\omega$ for a weak nonlinear partial differential equation of mixed type with fractional Hilfer operator $D^{\alpha,\gamma}$ in a positive rectangular domain $\Omega_1 = \{0 < t < b,\ 0 < x, y < l\}$ and with spectral parameter $\omega$ in a negative rectangular domain $\Omega_2 = \{-a < t < 0,\ 0 < x, y < l\}$.

The set of positive solutions $\Im$ of trigonometric Equation (33) with respect to spectral parameter $\omega$ was called a set of irregular values of the spectral parameter $\omega$. The set of the remaining values of the spectral parameter $\aleph = (0; \infty) \setminus \Im$ was called a set of regular values of the spectral parameter $\omega$.

For all regular values of the spectral parameter $\omega$ the quantity $\Delta_{n,m}(\omega)$ was nonzero. So, for large $n$, $m$ the values of $\Delta_{n,m}(\omega)$ could not become quite small and there the problem of "small denominators" did not arise. Therefore, for regular values of the spectral parameter $\omega$ the quantity $\Delta_{n,m}(\omega)$ was separated from zero and we considered the questions of one value solvability of the considering boundary value problems (1)–(5).

We studied the boundary value problem $T_\omega$ with following assumptions:

$$\varphi(x,y) \in C^3[0;l]^2, \ \varphi_{xxxx}(x,y) \in L_2[0;l]^2, \varphi_{yyyy}(x,y) \in L_2[0;l]^2;$$

$$f_i(x,y,u) \in C_{x,y,u}^{3,3,0}\left([0;l]^2 \times \mathbb{R}\right), f_{ixxxx}(x,y,u) \in L_2\left([0;l]^2 \times \mathbb{R}\right),$$

$$f_{iyyyy}(x,y,u) \in L_2\left([0;l]^2 \times \mathbb{R}\right);$$

$$\chi_{11} = \max_{i=\overline{1,3}} \max_{n,m \in \mathbb{N}} \max_{t \in [0;b]} \left| t^{1-\gamma} \eta_{inm}(t,\omega) \right| < \infty; \ \chi_{21} = \max_{i=\overline{1,3}} \max_{n,m \in \mathbb{N}} \max_{t \in [-a;0]} \left| \xi_{inm}(t,\omega) \right| < \infty;$$

$$\chi_{30} = \left\| \varphi_{xxxxyyyy}(x,y) \right\|_{L_2[0;l]^2} < \infty; \ \chi_{3i} = \left\| f_{ixxxxyyyy}(x,y,\gamma) \right\|_{L_2[0;l]^2} < \infty;$$

$$\left| f_{ixxxxyyyy}(x,y,\gamma_1) - f_{ixxxxyyyy}(x,y,\gamma_2) \right| \leq K_i(x,y) \left| \gamma_1 - \gamma_2 \right|,$$

$$K_{0i} = \left\| K_i(x,y) \right\|_{L_2[0;l]^2} < \infty;$$

$$\left| \Theta_i(\xi,x,y,u_1) - \Theta_i(\xi,x,y,u_2) \right| \leq \Theta_{1i}(x,y) \left| u_1 - u_2 \right|,$$

$$\Theta_{2i} = \left\| \Theta_{1i}(x,y) \right\|_{L_2[0;l]^2} < \infty, \ i = 1, 2;$$

$$\rho = \gamma_2 (\gamma_1 + \gamma_3) \gamma_4 < 1, \ \gamma_4 = \max\{b K_{01} \Theta_{21}; a K_{02} \Theta_{22}\}.$$

If these conditions were fulfilled, then the boundary value problem $T_\omega$ was uniquely solvable for regular values of the spectral parameter $\omega \in \aleph$ with these solutions represented in the form of the Fourier series (53) and (54) in the domains $\Omega_1$ and $\Omega_2$, respectively. There the series (53), (54) and (57)–(62) were convergent absolutely and uniformly in the corresponding domains $\Omega_1$ or $\Omega_2$.

For irregular values of the spectral parameter $\omega \in \Im$ and for some $k$, $s = k_1, ..., k_s$ the problem $T_\omega$ had an infinite number of solutions in the form of series (64) and (65), if there the condition (63) was fulfilled.

**Author Contributions:** Conceptualization, T.K.Y. and B.J.K. All authors have read and agreed to the published version of the manuscript.

**Funding:** This research received no external funding.

**Conflicts of Interest:** The author declares no conflicts of interest.

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
