# Peer review of "Boundary Value Problem for Weak Nonlinear Partial Differential Equations of Mixed Type with Fractional Hilfer Operator"

_axioms, doi:10.3390/axioms9020068_

Round 1

Reviewer 1 Report

Dear Respected Editor,

I have read this paper in details. Authors, in this paper, have considered a boundary value problem for a nonlinear partial differential equation of mixed type with Hilfer operator of fractional integro-differentiation in a positive rectangular domain and with spectral parameter in a negative rectangular domain.  They have constructed the solutions of nonlinear boundary value problems in the form of a Fourier series by using the Fourier series method. They have proved theorems on the existence and uniqueness of classical solution of the problem for regular values of the spectral parameter. For irregular values of the spectral parameter, an infinite number of solutions of the mixed equation in the form of a Fourier series are constructed by them.

1.Similarity index is of 27% for this paper. This number is  not too much for this paper. But, this number must be reduced a little more. 

2.At the end of all equations must be putted "COMMA" or "POINT" according to the typing rules. Therefore, they need to pre-check all the paper.

3.Conclusion section is too short. It must be extended a little more by giving some important properties of results.

4.The authors are request to add more details regarding their original contributions in this manuscript.

5.Reference section is too short. Moreover, some of papers cited in referece section are very old. They need to consider more papers and more comparision with their paper given as follows: Chaos, Solitons & Fractals 117 (2018): Applied Mathematics and Nonlinear Sciences,4(2), 289–304, 2019.; 16-20; Results in Physics, 15 (102555), 1-7, 2019; The European Physical Journal Plus 134.9 (2019): 429.; . New and extended applications of the natural and Sumudu transforms: Fractional diffusion and Stokes fluid flow realms. In Advances in Real and Complex Analysis with Applications (pp. 107-120). Birkhäuser, Singapore.; International Journal of Fluid Mechanics Research, vol. 30, (5) (2003), 463-472.; Modern Physics Letters B, 34(3), 2050034 (18 pages), 2020; Journal of Advanced Engineering and Computation, vol. 2 (4), (2018), pp. 224-238.; International Applied Mechanics, vol. 39 (10), (2003), 1115-1145.; Applied Mathematics and Nonlinear Sciences, 4(1),  35–42, 2019.; Thermal Science, 19(3), 959-966, 2015.; Chaos, Solitons & Fractals 136, 109812, 2020; Physica A: Statistical Mechanics and its Applications 542, 123516, 2020; Applied Mathematics and Computation 316 (2018), 504-515. Open Mathematics, 13(1), 547–556, 2015,;

6.Papers cited in references section must be rewritten according to journal style before further process.

Briefly, after these modifications, this paper may be reconsidered for the further steps on the way of publishing.
With my best regards

Author Response

Many thanks to you dear referee for the good support of my work and for the valuable advice that undoubtedly contributed to improving the quality of the work.

I made the following changes to my article:

  1. Wrote an introduction to the article; 2. Wrote (expanded) the conclusion to the article; 3). I add (expanded the list of references) new references.

Sincerely yours Tursun Yuldashev

Reviewer 2 Report

The Authors consider a boundary value problem with Hilfer operator. The material presented in the manuscript is of interest for scholars in the field of fractional calculus and related fields. The presentation in the manuscript must be improved, and the new results in the manuscript which are different than those in previous works must be clearly elaborated.

Therefore, I list some comments and suggestions on the manuscript.

Title: The title is very general. It should be more specific in order to reflect the content of the manuscript.

The Authors should add a section “Introduction” where will explain the importance of fractional equations and fractional operators, related to the problem considered in the manuscript, what has been done so far, etc.

Definition of Hilfer operator on page 1: The Authors should add appropriate citation where this operator was introduced by R. Hilfer.

Please explain when Hilfer operator corresponds to the Caputo derivative and when to Riemann-Liouville derivative.

After introduction of the Hilfer operator the Authors should add explanation of its importance, applications, etc.

There are many applications of this operator. For example, Hilfer showed that time fractional derivatives are equivalent to infinitesimal generators of generalized time fractional evolution, which arises in the transition from microscopic to macroscopic time scales; see for example [R. Hilfer, Chem. Phys. 284, 399 (2002); R. Hilfer, Fractals 11, 251 (2003)].

Hilfer also showed that this transition from ordinary time derivative to fractional time derivative arises in physical problems; see for example [R. Hilfer, Application of Fractional Calculus in Physics (World Scientific Publishing Company, Singapore) 2000].

The Hilfer’s idea on time fractional evolution is presented in Chapter 9 of the book [J. Klafter, S.C. Lim and R. Metzler, Fractional Dynamics, Recent Advances (World Scientific, Singapore) 2011] as well, and in some recent papers on Hilfer-composite time fractional derivative. For example, fractional diffusion equations with Hilfer operator were analyzed in [J. Phys. A: Math. Theor. 44, 255203 (2011)], [Physica A 391, 2527 (2012)], [Appl. Math. Comput. 242, 576 (2014)], [Fractional Equations and Models: Theory and Applications, Springer Nature, 2019] and related works.

Before Eq. (35): there should be t^{\gamma-1} instead of t^{\alpha-1}.

The Conclusions should be reformulated. One can not see what has been done in the manuscript just with referring to particular equations from the manuscript.

Author Response

(The authors gave the same response as above.)

Round 2

Reviewer 2 Report

The Authors have improved this revised version of the manuscript. However, the Introduction is very short and with limited information. In my previous report I just mention some possible applications of the Hilfer derivative and its importance, so the Authors should rephrase my points in the report, not just copy-paste the text.

Here are some work which authors should consult in order to improve the Introduction.

[1] Applications of Fractional Calculus in Physics (World scientific, 2000)

[2] Experimental evidence for fractional time evolution in glass forming materials, Chemical Physics 284, 399-408 (2002)

[3] On fractional relaxation, Fractals 11, 251-257 (2003)

[4] Fractional diffusion equation with a generalized Riemann–Liouville time fractional derivative, Journal of Physics A: Mathematical and Theoretical 44 (25), 255203 (2011)

[5] Generalized space-time fractional diffusion equation with composite fractional time derivative, Physica A 391 (8), 2527-2542 (2012)

[6] Hilfer-Prabhakar derivatives and some applications, Applied Mathematics and Computation 242, 576-589 (2014)

[7] Fractional Equations and Models: Theory and Applications (Springer, 2019), see Chapter II and Chapter IV

...etc.

After this the manuscript could be suitable for publication in Axioms.

Author Response

Thankyou for your supporting me! I made an extension to the introductory part of the article.